# Morphological Alterations of Conal Ridges and Differential Expression of AP2α in the Offspring Hearts of Experimental Diabetic Rats

**DOI:** 10.3390/ijms26115061

**Published:** 2025-05-24

**Authors:** Tania Cristina Ramírez-Fuentes, Ricardo Jaime-Cruz, Carlos César Patiño-Morales, Laura Villavicencio-Guzmán, Juan Carlos Corona, María Cristina Revilla-Monsalve, Rosa Adriana Jarillo-Luna, Marcela Salazar-García

**Affiliations:** 1Research Laboratory of Developmental Biology and Experimental Teratogenesis, Children’s Hospital of México Federico Gomez, Mexico City 06720, Mexico; tania.ramirez3190@outlook.com (T.C.R.-F.); ricardo.jaime.cruz@gmail.com (R.J.-C.); cpatino@cua.uam.mx (C.C.P.-M.); villagu@yahoo.com (L.V.-G.); 2Sección de Estudios de Posgrado e Investigación, Escuela Superior de Medicina del Instituto Politécnico Nacional, Mexico City 11340, Mexico; 3Departamento de Ciencias de la Salud, Universidad Tecnológica de México-UNITEC México-Campus Sur, Mexico City 09810, Mexico; 4Facultad de Medicina, Universidad Nacional Autónoma de México, Mexico City 04360, Mexico; 5Laboratory of Neurosciences, Hospital Infantil de México Federico Gómez, Mexico City 06720, Mexico; jcorona@himfg.edu.mx; 6Unidad de Investigación en Enfermedades Metabólicas, Centro Médico Nacional Siglo XXI, Instituto Mexicano del Seguro Social, Mexico City 06720, Mexico; macrisrev@gmail.com

**Keywords:** hyperglycemia, heart development, conotruncal cardiopathies, conal ridges, cardiac outflow tracts

## Abstract

Neural crest cells (NCCs) play a significant role in the development of ventricular outflow tracts (OFTs), and cardiac neural crest cells (cNCCs) are involved in the development of the embryonic conus, suggesting that these cell lineages may be a teratogenic target for the development of cardiopathies in offspring conceived under a hyperglycemic environment. We evaluate the effect of the hyperglycemic intrauterine environment on the morphological and anatomical changes in the conal ridges along with the alterations in the spatiotemporal expression of AP2α in offspring hearts at 13, 15, and 17 DPC. The anatomical and histological analysis of the hearts in the experimental group presented smaller dimensions compared to the control group in the offspring at the three ages studied. Consequently, this resulted in a hyperglycemic environment that altered the immunostaining of AP2α in the hearts of the offspring at the three ages studied. Thus, the hyperglycemic intrauterine environment in offspring caused important morphological alterations in the development of conal ridges and promoted the generation of conotruncal heart defects in which the double outlet of the right ventricle, the atrioventricular (AV) canal, predominated. Therefore, knowing that exposing the offspring to more glucose potentially can lead to complications during organogenesis of the circulatory and central nervous systems.

## 1. Introduction

For many decades, the embryonic development of the ventricular outflow tracts (OFTs) has been a controversial topic. Formerly, it was proposed that the embryonic origin of both OFTs is the embryonic cone, with the posterior region giving rise to the left ventricular OFT and the anterior region, the right ventricular OFT [1]. However, lineage-tracing studies in chick embryos in vivo have challenged this view, suggesting that part of the trabeculated free wall of the right ventricle (RV) originates from the myocardium that initially belonged to the embryonic segment known as the proximal OFT or conus by several authors [2]. According to this information, in vivo labeling was also used in the same animal model to trace the destination of the conal walls of the embryonic OFT, and it was found that these conal walls are incorporated into the RV and its OFT [3]. Correlating this information, myocardialization of the conal crests after their fusion has been reported in rats at gestational day 10 [4]. Regarding the development of the arterial pole, the classic literature considers that the truncus, which is the distal segment of the embryonic OFT of the heart, gives rise to the aortic and pulmonary sigmoid cusps, as well as participates in the formation of the distal portion of the ventricular OFTs and the proximal portion of the great arteries [5]. However, histological research has demonstrated that the proximal region of the pulmonary and aortic arteries does not give rise from the embryonic truncus but from the aortic sac [6,7]. It is important to highlight that cardiac neural crest cells (cNCCs) play a central role in the development of the cardiac OFT, sigmoid valves, and great arteries. The cNCCs migrate from the pharyngeal arches at the level of the occipital somites one to three to the aorticopulmonary septum. Experimental studies have shown that when this cell lineage is removed bilaterally prior to migration, congenital heart defects, such as the common arterial trunk or transposition of the great arteries, result [8]. To expand this information, cNCCs migrating to the third, fourth, and sixth pharyngeal arches have been identified as invading the heart through the wall of the developing OFT. These findings have provided a guideline for the research into the participation of NCCs in cardiac development and the molecular processes involved; on this matter, the transcription factor activating protein (AP2α), a regulator of NCC differentiation and segregation, has been involved in the morphogenesis of the cardiac OFT. AP2α-deficient embryos predominantly exhibit double outflow of the right ventricle [9].

Approximately half of the congenital heart defects observed in children born to mothers with hyperglycemia during pregnancy are conotruncal anomalies [10,11]. Induced hyperglycemia in pregnant rats has been reported to chronologically dysregulate the intrauterine development of the offspring, restricting embryo–fetal growth and delaying the remodeling and maturation phases of structures derived from NCCs, leading to craniofacial abnormalities as well as neurobehavioral and hippocampal alterations [12,13]. In HH 8–12 chick embryo NCCs exposed to high glucose concentrations, an increase in cardiac malformations mostly located in the OFT and large arteries has been observed, reaffirming the vulnerability of NCCs to hyperglycemia [14]. The frequency of cardiac anomalies in the children of mothers experiencing hyperglycemia in the early weeks of gestation is higher because the heart, in the pre-septation stage, is highly susceptible to teratogenic factors that reduce cellular proliferation and migration in certain embryonic tissues [15]. Early impairment can lead to severe heart disease due to defects in the endocardial cushions and myofibrillar aberrations. Recently, it was observed that the atrioventricular (AV) canal cushions did not fuse in embryos subjected to cNCC ablation, affecting the normal septation process of the AV canal and altering blood flow; however, the size of the cushions was not affected [16]. Current evidence suggests that maternal hyperglycemia has direct and indirect effects through reactive oxygen species on cardiac morphogenesis and histodifferentiation, impacting processes such as cell proliferation, apoptosis, and migration [17]. Therefore, the present study was designed to determine the morphological and histological changes in conotruncal ridges under hyperglycemic environment, the changes in AP2α as a marker of cNCCs, and the incidence of conotruncal heart defects in the offspring of diabetic and control rats at three different stages.

## 2. Results

### 2.1. Glucose Levels of Rats with STZ-Induced Diabetes

Blood glucose levels were recorded every 48 h, showing a significant difference between the glucose levels in the control and the experimental group (Figure 1B). In the pregnant rats of the experimental group, the values ranged from 319 mg/dL to 600 mg/dL. In contrast, glucose levels in the control rats remained between 86 and 155 mg/dL until euthanasia. These results validate the experimental diabetic rat model used in this study. Experimental pregnant rats had smaller litters (an average of 7 offspring per litter), with a higher number of resorptions and deaths, whereas control rats had an average of 13 offspring per litter.

### 2.2. Anatomical and Histological Changes of Conal Ridges in the Offspring of Diabetic Rats

To determine hyperglycemia-induced anatomical and histological changes in the offspring’s hearts at 13, 15, and 17 days post-conception (DPC), the offspring of the experimental group were analyzed at these three developmental stages. Figure 2A presents a micrograph of a control group heart at 13 DPC (equivalent to 30 ± 1 days of gestation in humans), with the following reference points for measurements: (1) OFT length, (2) AV canal length, (3) major diameter, and (4) minor diameter. Embryonic rat heart measurements are summarized in Table 1.

In embryos from the control group at 13 DPC, the embryonic OFT was observed to be anterior and central, with the primitive atria positioned above their respective ventricles (Figure 2B). This stage corresponds to the end of the process of torsion and looping of the cardiac tube. However, in embryos developed in a hyperglycemic environment (experimental group), the embryonic OFT appears displaced to the right, parallel to the AV canal, leaving the primitive atrial region more towards the left (Figure 2B). A difference was observed in the length of the major diameter (from the AV canal groove to the cardiac apex) in the AV canal and the embryonic OFT in the experimental group. In the images from hematoxylin/eosin (H/E)-stained tissue sections at the level of the proximal embryonic OFT in the control group, two protrusions—one anterior and one posterior—were observed. These protrusions consisted of regularly shaped and continuous mesenchyme, bordered by simple squamous epithelium, corresponding to the conal ridges, which were closely situated but not fused. Myocardial tissue in this region appeared continuous, with a consistent thickness throughout the entire perimeter of the embryonic OFT (Figure 2C). The experimental group showed two conal ridges side by side, with the most pronounced changes observed in the sinistroventral ridge (SVR). In 55% of the analyzed sections, the mesenchyme appeared discontinuous, in contrast to the control group. In addition, the endocardium adjacent to this area showed squamous cell compaction and the presence of cuboidal cells. Regarding the myocardial tissue, cells with an ovoid shape, rounded nucleus, and abundant cytoplasm were observed, with irregular thickness along the circumference of the proximal embryonic OFT (Figure 2C).

At 15 DPC (36 ± 1 days of gestation in humans), hearts of the experimental embryos exhibited atrial hypoplasia, dilation of the trabeculated region of the RV, and widening of the embryonic OFT (Figure 2D). A difference in the major diameter, the minor diameter, and both AV canal and embryonic OFT length was observed in the experimental group. Histologically, the hearts in the control group exhibited the embryonic OFT localized to the left, consistent with the findings from the anatomical analysis. The SVR of the conus was observed to be continuous with the ventrosuperior cushion of the AV canal, and both conal ridges were seen to be fused. In the myocardium, the formation of incipient trabeculae was observed in the RV-free wall (Figure 2E). In the experimental group, the embryonic OFT was not observed to be localized to the left, and the conal ridges were fused to form an anterior and a posterior cavity. Myocardial tissue showed reduced thickness, with sparse, short, and thin ventricular trabeculae (Figure 2E). 

At 17 DPC (38 ± 1 weeks of gestation in humans), differences in dimensions and development were markedly reduced in the experimental group compared to controls (Figure 2F). A total of 80% of hearts from the experimental group at 17 DPC exposed to hyperglycemia exhibited AV canal concordance, with hypoplastic atria and apparently smoother walls. The RV was larger and bulging in shape, while the left ventricle was hypoplastic compared to those in the control group (Figure 2F). In the control group a situs solitus, AV canal and ventriculoarterial concordance, a pulmonary infundibulum with muscular walls, an aortic vestibule with a muscular portion represented by the IVS and a fibrous portion due to mitroaortic continuity, and ventricles with well-defined and formed trabeculae were observed (Figure 2G).

The effects of hyperglycemia on microscopic and macroscopic alterations in the offspring heart were further evaluated at 13, 15, and 17 DPC. As detailed in Table 2, microscopic and macroscopic analyses revealed the following alterations in the experimental group: side-by-side arteries (nine hearts), double outflow of the right ventricle (seven hearts), persistent AV canal (seven hearts), pulmonary artery (PA) stenosis (five hearts), right ventricle dilation (five hearts) and normal morphology (three hearts, matching observations in the control group).

### 2.3. Alterations in the Expression of AP2α in the Heart of the Offspring of Diabetic Rats

The effects of the hyperglycemic intrauterine environment on AP2α expression in the offspring hearts were evaluated at 13, 15, and 17 DPC. Analysis of hearts at 13 DPC revealed similar AP2α expression in the dextrodorsal ridge (DDR) myocardium between the control and experimental groups (Figure 3A,B), while in the DDR mesenchyme, a higher expression of the AP2α was observed in the control group, but it was not significant compared with the experimental group (Figure 3A,B). In the SVR myocardium, a statistically significant higher expression of the AP2α was observed in the control group compared with the experimental group (Figure 3A,B), while in the SVR mesenchyme, the expression of the AP2α was similar in the control group compared with the experimental group (Figure 3A,B).

In the myocardium of offspring at 15 DPC, the expression of AP2α in the DDR was not significant in the control group compared with the experimental group (Figure 3C,D), while in the SVR, there was no difference between the control and experimental groups (Figure 3C,D). In the mesenchyme, the expression of AP2α in the DDR was higher in the control group compared with the experimental group, but it was not significant (Figure 3C,D). While in the SVR, the expression of AP2α was similar in the control group compared with the experimental group (Figure 3C,D).

Finally, at 17 DPC, the walls of the supraventricular crest, derived from the fusion of the conal ridges with myocardial characteristics, showed a statistically significant increase in the expression of AP2α in the control group compared with the experimental group (Figure 3E,F).

## 3. Discussion

Normal heart development remains controversial, complicating the understanding of the pathophysiology of congenital heart diseases, which have a higher incidence when the mother experiences hyperglycemia during gestation [18,19]. Thus, in pregestational diabetes, excessive glucose can cross the blood–placental barrier, exposing the fetus to more glucose and potentially leading to several complications during fetal development, such as the organogenesis of the heart and brain, increasing the susceptibility of offspring to develop cardiometabolic diseases later in life due to epigenetic changes during fetal development [20,21]. Our findings, both macroscopically and histologically, revealed the presence of complex conotruncal heart defects due to teratogenic effects during the critical period of cardiovascular development, which have been reported in children of diabetic mothers as part of the diabetic fetopathy syndrome [22]. Additionally, a reduction in heart size and development was observed, consistent with data from other researchers who reported that induced hyperglycemia in pregnant rats and other biological models chronologically dysregulates progeny development, restricting embryo–fetal growth and delaying the remodeling and maturation of structures derived from NCCs [12,23]. In addition, gestational diabetes could induce structural birth defects, including neural tube defects and congenital heart defects in human fetuses. Hyperglycemia of maternal diabetes causes oxidative stress in the developing neuroepithelium and the embryonic heart, leading to the activation of pro-apoptotic kinases and cell death [17,24,25].

In the myocardial tissue of the offspring from diabetic rats, we observed underdeveloped cardiomyocytes, thin myocardial walls, and irregular trabeculae. This is consistent with previous studies showing that glucose dose-dependently suppresses the maturation of cardiomyocyte cellular architecture [26] and leads to a decrease in the number of cardiomyocytes, as well as delays in the process of delamination and compaction of ventricular trabeculae [23]. However, in our study, it is remarkable to point out the modifications found in the mesenchyme of the conal ridges, which are found with less cellularity, and the SVR presented greater affectation in its sculpting due to the abnormal development of a groove that causes a discontinuity in the periphery of the ridge that faces the conal lumen. In the past, it was thought that both conal crests fuse to form an anterior cone or infundibular primordium (IP) of the RV and a posterior cone or aortic vestibule primordium [2].

However, an in vivo labeling study in chick embryos has questioned such information, evidencing that the embryonic conus is exclusively incorporated into the RV [3,4]. Furthermore, histological analyses have shown that both conal ridges fuse to form the supraventricular crest, which belongs to the IP, and not to the aorto-pulmonary septum [4]. These findings align with the results of this study, suggesting that defective fusion of the conal ridges may lead to malformations primarily localized to the RV, such as those observed in hearts exposed to a hyperglycemic intrauterine environment.

The expression pattern of AP2α was higher in the myocardial tissue of the control group, consistent with previous studies demonstrating cNCC interaction with the second heart field (SHF) [27], which participates in outflow tract (OFT) myocardial formation and contributes to myocardial development, with persistence lasting up to two weeks after birth [28]. Experimental findings indicate that a high glucose concentration inhibits the development of cranial NCCs in vitro [29], from which cNCCs are derived. In the experimental group, increased AP2α expression was observed in the myocardium at 15 DPC hearts, suggesting a higher involvement of cNCC. These findings align with the developmental delay noted in morphological analyses and corroborate earlier reports [12]. According to previous studies, a relevant finding is that this expression is more evident in the myocardium of the DDR, whose wall will be incorporated into the RV [3,4], supporting the RV dilatation and the alterations in the ventricular trabeculae found in our experiments, which is extremely important because errors in the maturation of ventricular structures could affect the contractile capacity of the RV. As mentioned above, both ridges fuse to form the supraventricular crest with myocardial characteristics, which is a tissue that could have differentiated from the NCC [30]. Control group specimens exhibited elevated AP2α expression in the supraventricular crest at 17 DPC. In contrast, experimental group hearts displayed anomalous fusion of the conal ridges alongside reduced AP2α expression, suggesting diminished neural crest cell (NCC) involvement. These findings implicate impaired NCC contribution in the pathogenesis of double outlet right ventricle (DORV). 

## 4. Materials and Methods

### 4.1. Animals

We used 27 Sprague-Dawley female juvenile rats of at least 10 weeks of age with weights between 250 and 270 g, 15 for the experimental group and 12 for the control group. These animals were maintained on a 12-h light/dark cycle and fed with LabDiet 5008 pellets (27% protein, 16% fat, and 65% carbohydrates) and water ad libitum. The estrous or proestrous phase was identified, mating 2 female rats with a male. Vaginal swabs were subsequently collected, smeared onto slides, and examined microscopically to detect sperm presence and confirm pregnancy onset. Rats with positive sperm identification were randomly assigned to control or experimental groups. All animal use protocols and research procedures strictly adhered to the Mexican Official Standard (NOM-062-ZOO-1999) [31] guidelines for the care and use of laboratory animals. These protocols were approved by the ethical and research committees of our institution (HIM/2021/046, HIM/2019/057, and HIM/2018/046).

### 4.2. Induction of Hyperglycemia in Rats

Diabetes was induced following protocols described in previous research [11,29]. On the fifth day of pregnancy, a single intraperitoneal dose of streptozotocin (STZ; 50 mg/kg body weight, dissolved in citrate buffer, Sigma-Aldrich, St. Louis, MO, USA) was administered. To confirm diabetic status, random blood glucose levels ≥200 mg/dL were verified 48 h post-induction in the pregnant rats. Both the control (non-diabetic) and hyperglycemic (experimental) groups were monitored for weight and blood glucose levels every 48 h during gestation. Glucose measurements were performed via capillary blood sampling using a glucometer and test strips (FreeStyle Optium Neo, Abbott Diabetes Care, Inc., Chicago, IL, USA).

### 4.3. Obtaining Embryos and Fetuses of Rats with STZ-Induced Diabetes

Pregnant control and diabetic rats were assigned to three subgroups to collect embryos or fetuses at 13, 15, and 17 DPC. These timepoints correspond to stages of embryonic outflow tract (OFT) conal ridge development. Following euthanasia with a lethal dose of pentobarbital, laparotomy was performed to remove the uterine horns. Using microdissection under a stereoscopic microscope (Leica M80, Leica Microsystems), the gestational sacs were carefully opened to extract embryos or fetuses (n = 20 per subgroup).

### 4.4. Anatomical and Histological Analysis

The hearts of the collected embryos and fetuses were extracted and photographed to document their external morphology using a stereomicroscope (Zeiss Lumar V12) equipped with a digital camera (Axiovision LE 4.8 software). Subsequently, a sequential segmental anatomical analysis of the atrial, ventricular, and OFT regions was performed. Morphometric parameters were measured, including the major axis (from the AV canal groove to the cardiac apex), the minor axis (maximum transverse ventricular diameter), the length of the AV canal, and the embryonic OFT. Data were analyzed by group (control vs. experimental) using Student’s *t*-test. For histopathological evaluation, the hearts were fixed in 4% paraformaldehyde in PBS (pH 7.2–7.4). Following fixation, specimens were dehydrated, cleared, and embedded in paraffin. Transverse tissue sections (5 µm thickness) were prepared and stained using the hematoxylin and eosin (H/E) technique. Finally, the stained sections were analyzed to assess the morphological features of the conal ridge mesenchyme and the myocardial tissue of the proximal embryonic OFT. Cardiomyopathy was confirmed through histological observations in the transverse plane, focusing on structural anomalies in the myocardial architecture and mesenchymal organization.

### 4.5. Immunofluorescence of AP2α

To determine the cNCC presence in conal ridges, immunodetection of AP2α (3B5) (sc12726, Santa Cruz Biotechnology, Dallas, TX, USA) was performed (1:150 dilution) by indirect immunofluorescence. Under semi-dark conditions, goat anti-mouse IgG-FITC secondary antibody (sc-2010, Santa Cruz Biotechnology, Dallas, TX, USA) (1:100 dilution) was incubated for 4 hours. Nuclei were counterstained with RedDot™ (Biotium, Fremont, CA, USA; 1:150 dilution). Slides were imaged using a confocal microscope (Carl Zeiss, Oberkochen, Germany), and microphotographs were acquired with ZEN 2010 software (Carl Zeiss, Germany). Fluorescence intensity was quantified using ImageJ software (v.2.0, NIH, Bethesda, MD, USA) [32]. The experimental workflow is illustrated in the schematic diagram in Figure 1A.

### 4.6. Statistical Analysis

Statistical analysis was performed using GraphPad Prism Software (Version 8.01, GraphPad, Inc., La Jolla, CA, USA). Data are expressed as the mean ± standard error of the mean (SEM) from at least eight independent experiments. Maternal metabolic and morphometric parameters from control and experimental groups were compared using two-tailed unpaired Student’s *t*-tests. Differences in AP2α immunostaining between groups were analyzed by one-way ANOVA followed by Tukey’s post hoc test. Statistical significance was defined as *p* < 0.05.

## 5. Conclusions

The findings demonstrate that hyperglycemia disrupts embryonic heart development and gross cardiac morphology in rats, with cNCCs playing a critical role in conal ridge formation. These observations suggest that the embryonic conus may represent a teratogenic target for hyperglycemia-induced congenital heart defects in offspring exposed to maternal hyperglycemia in utero. Furthermore, the rat model proves valuable for investigating the pathogenesis of congenital heart defects associated with gestational diabetes and underscores the risks of poor glycemic control during pregnancy. Excessive glucose traversing the placental barrier elevates fetal glucose exposure, which may perturb organogenesis of the cardiovascular and central nervous systems.

## Figures and Tables

**Figure 1 ijms-26-05061-f001:**
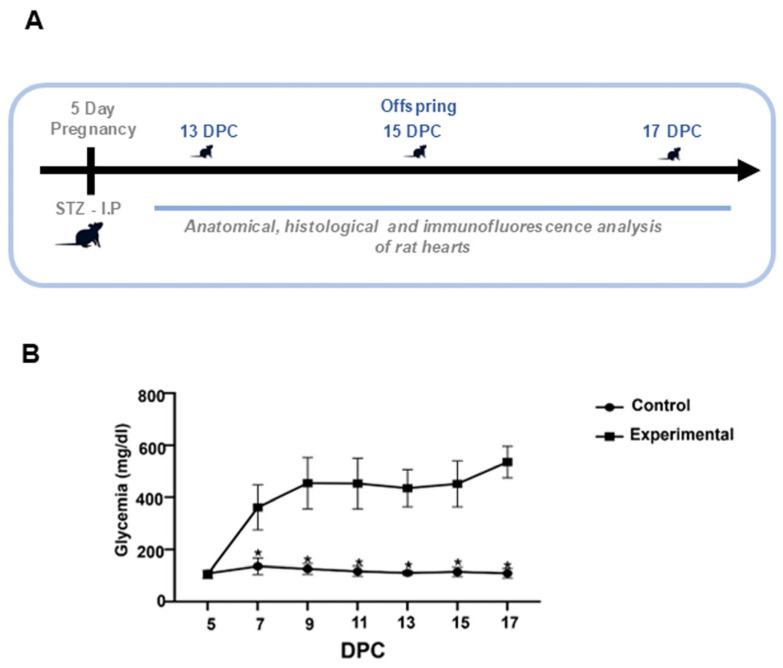
Maternal glycemic levels in rats STZ-induced diabetes. (**A**) Schematic representation of experimental procedures and timeline. (**B**) Blood glucose in pregnant rats is represented in milligrams per deciliter (mg/dL). Experimental rats exhibited sustained hyperglycemia, reaching and maintaining levels > 200 mg/dL. Data are presented as mean ± SEM (n = 7 rats per group). A significant difference between the control and experimental group is indicated * *p* < 0.05 by a two-tailed unpaired Student’s *t*-test.

**Figure 2 ijms-26-05061-f002:**
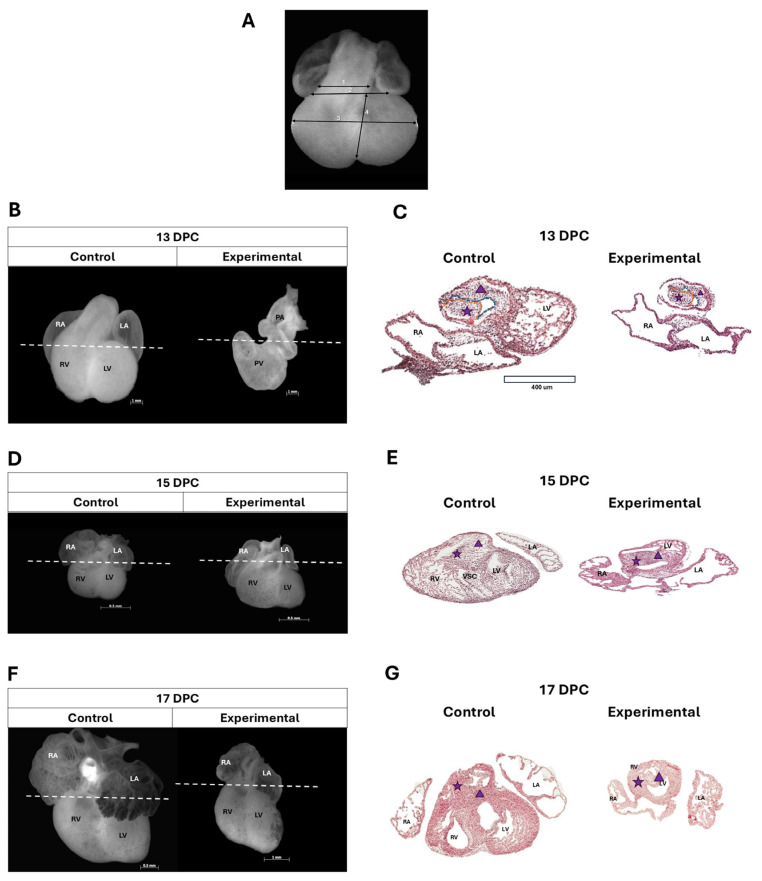
Anatomical and histological changes in the heart of the offspring of diabetic rats. (**A**) Macroscopic reference points for heart measurements: 1: outflow tract (OFT) length; 2: AV canal length; 3: major diameter; and 4: minor diameter. (**B**) Macroscopic analysis of experimental group hearts at 13 DPC revealed reduced dimensions, rightward displacement of the OFT (parallel to the AV canal), and leftward positioning of the primitive atrial region. (**C**) Representative images from hematoxylin/eosin (H/E)-stained tissue sections from the experimental group at 13 DPC showed a decrease in cellularity in both conal ridges with changes in the sinistroventral ridge (SVR) morphology, which presents a cleft at 13 DPC, shown on the blue dotted line and in the myocardial tissue, cells with an ovoid shape, rounded nucleus, and abundant cytoplasm, was observed. (**D**) The macroscopic analysis of the heart in the experimental group at 15 DPC presented atrial hypoplasia, dilation of the trabeculated region of the RV, and widened the embryonic OFT. (**E**) showed mislocalization of the embryonic OFT (non-leftward orientation), fused conal ridges, and myocardial tissue with reduced thickness. Ventricular trabeculae were sparse, short, and thin. (**F**) At 17 DPC, the macroscopic analysis of the heart in the experimental group presented smaller dimensions in development and presented the AV concordance with hypoplastic atria and smoother walls, and the RV was larger and bulging in shape while the left ventricle was hypoplastic. (**G**) The stained tissue sections in the experimental group at 17 DPC showed that OFT had poor rotation, lateral fusion, and malformations due to impaired OFT formation. Scale bars, 400 µm. White dashed lines represent the cutting level; the purple star represents dextrodorsal ridge, which is delimited by the red line; the purple triangle represents sinistroventral ridge, which is delimited by blue line. Left atrium (LA); right atrium (RA); right ventricle (RV); left ventricle (LV); pulmonary vein (PV); pulmonary artery (PA); and (VSC) ventrosuperior cushion.

**Figure 3 ijms-26-05061-f003:**
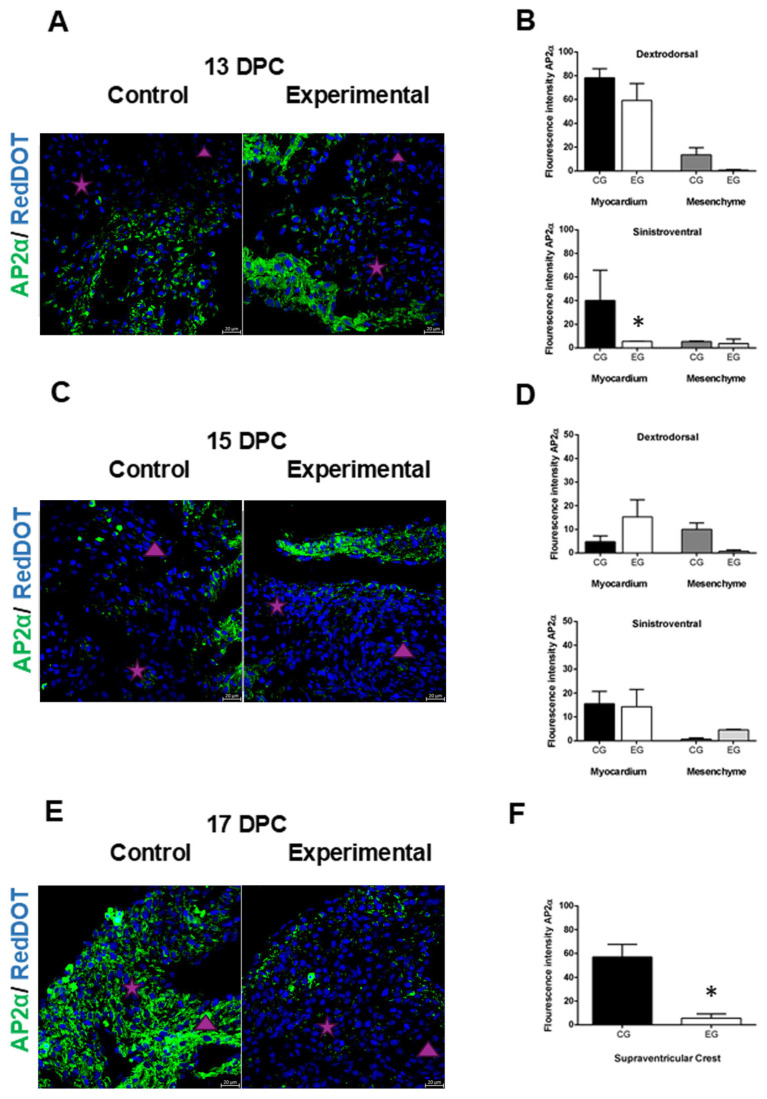
Changes in AP2α expression in offspring hearts of diabetic rats. (**A**,**C**,**E**). Immunostaining of AP2α (green) as an NCC marker was higher in myocardial tissue in both conal ridges in the control group, while in the hearts of the experimental group, the highest expression was in the myocardium at 13 and 15 DPC, which was decreased at 17 DPC. (**B**) Quantitative analysis of AP2α fluorescence intensity at 13 DPC showed no statistically significant difference in dextrodorsal ridge (DDR) myocardium compared to controls, while in the sinistroventral (SVR) myocardium, a statistically significant difference was observed compared with the control group. In the DDR and SVR mesenchyme, no statistically significant difference was observed compared with the control group. (**D**) Quantitative analysis of the fluorescence intensity of AP2α in the DDR myocardium or in the SVR myocardium at 15 DPC was not statistically significant compared with the control group. While in the DDR mesenchyme and SVR mesenchyme, neither a statistically significant increase in the expression of AP2α was observed compared with the control group. (**F**) Quantitative analysis of the fluorescence intensity of AP2α in the supraventricular crest at 17 DPC was statistically significant in the control group as compared with the experimental group. Statistical comparison was performed using one-way ANOVA with Tukey post hoc tests, * *p*  <  0.05 versus the control group, or two-tailed unpaired Student’s *t*-test. The purple star represents dextrodorsal ridge; the purple triangle represents sinistroventral ridge. Magnification 40×.

**Table 1 ijms-26-05061-t001:** Embryonic heart lengths in the offspring of diabetic rats.

Rat Heart Lengths (mm)								
	Control				Experimental			
DPC	OFT	AV	MD	mD	OFT	AV	MD	mD
**13**	0.5 ± 0.27	0.64 ± 0.3	1.36 ± 1.36	0.71 ± 0.34	0.32 ± 0.06 *	0.38 ± 0.1 *	0.81 ± 0.04 *	0.5 ± 0.04
**15**	0.58 ± 0.1	0.96 ± 0.12	1.68 ± 0.03	0.94 ± 0.07	0.56 ± 0.03 *	1.1 ± 0.09 *	1.66 ± 0.05 *	0.9 ± 0.08 *
**17**	0.54 ± 0.09	1.54 ± 0.17	2.34 ± 0.26	1.51 ± 0.21	0.54 ± 0.16 *	1.06 ± 0.32 *	1.83 ± 0.39 *	1.5 ± 0.19 *

All measurements were conducted in hearts at 13, 15, and 17 DPC from control and experimental groups. Data for these three embryonic stages are presented as mean ± SD. The following parameters were measured in the rat heart: OFT (outflow tract length), AV (atrioventricular canal length), MD (major diameter), and mD (minor diameter). Abbreviations used include: DPC (days post coitum); statistical significance (*); and mm (millimeter).

**Table 2 ijms-26-05061-t002:** Incidence of congenital cardiopathy.

Cardiopathy	Incidence
Side-by-side artery	9
Double outflow of the RV	7
AV canal	7
PA stenosis	5
RV dilation	5
Normal	3

Measurements were conducted in fetuses at 17 days post-conception (DPC) in the experimental group. Data are presented as the number of hearts exhibiting alterations (n = 20 rats per group).

## Data Availability

The original contributions presented in this study are included in the research article; any further inquiries can be directed to the corresponding authors.

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
