# Peer review of "Morphological Alterations of Conal Ridges and Differential Expression of AP2α in the Offspring Hearts of Experimental Diabetic Rats"

_ijms, 2025, doi:10.3390/ijms26115061_

Round 1
Reviewer 1 Report
Comments and Suggestions for Authors
Ramirez-Fuentes et al have studied defects in hearts following maternal hyperglycemia in a diabetic rat model. They reveal major structural defects in the myocardium, overall heart shape and in the outflow tract cushions. While the conclusions appear to be important and are clearly stated on page 11 there are a number of critical major points need to be taken into account.
- Is heart size specifically affected in experimental embryos? Or is overall embryo size reduced? If possible please add this information to Table 1. In this table, for clarity, please include the features measured directly in the table rather than numbers 1-4. These could also be directly indicated in panel 2A by adding the terms to the figure.
- The small heart size points to more than cNCC defects - this needs to be discussed. Indeed, expression of AP2a in the myocardial wall is not related to cNCC defects, as this is not derived from the neural crest but the second heart field. This could be linked with irregular thickness of OFT myocardium. Moreover, this point could also be better shown in high magnification images.
- Concerning Figure 2, please ensure that all the sections are in the same orientation.
- In Table 2 it is unclear how can there be an incidence of 22.2% if 20 hearts were scored per group. It would be simpler to give the actual numbers of hearts scored. It would also significantly improve the manuscript if the authors could show examples of these different 17 DPC defects in an additional figure.
- In Figure 3 please label the microscopy panels so it is clear what the reader is seeing. Low magnification views would be helpful here. Scale bars must be added to the micrographs as opposed to mentioning the magnification in the legend (as in Figure 2).
- Quantification of immunofluorescence is very challenging, please clarify in detail how this was performed. Did the authors normalize by quantifying a second reference protein?
- Please comment on lower expression in control embryos at some sites of AP2a expression. This is evident in panels 3A and 3C. Please indicate what these labelled structures are in the manuscript.
- The contribution of cNCC to the cardiac inflow is controversial - this point could potentially be removed (line 67).
- In the introduction, please provide more references to mammalian OFT development as many of the initial references are to the avian heart.
- The manuscript needs to be carefully re-read for English. Examples to be corrected are:
Line 46, Formerly, it was considered that the.... the posterior region of which would give rise to the...
Line 63 and when this cell lineage....arteries type results.
Line 66, cNCC migrating....has been identified,
line 117, Figure 2A shows
line 161 - and also main text on line 176, widening of the embryonic OFT
line 206, we evaluated
line 281, belonging to the IP
line 283, associated with the RV
- Typo on line 135 images
Comments on the Quality of English Language
- The manuscript needs to be carefully re-read for English. Examples to be corrected are:
Line 46, Formerly, it was considered that the.... the posterior region of which would give rise to the...
Line 63 and when this cell lineage....arteries type results.
Line 66, cNCC migrating....has been identified,
line 117, Figure 2A shows
line 161 - and also main text on line 176, widening of the embryonic OFT
line 206, we evaluated
line 281, belonging to the IP
line 283, associated with the RV
Reviewer 2 Report
Comments and Suggestions for Authors
Comments:
- How did the authors perform the vaginal smears? Briefly describe the procedure or cite the appropriate reference(s).
- How was the streptozotocin prepared?
- How was the blood obtained to verify blood glucose levels and classify the rats as diabetic? Were fasting blood samples used? If yes, how long was the fasting:
- How many female rats were used for the study, and how many male rats?
- When did the authors collect blood for glucose determination to confirm diabetes after administering streptozotocin?
- How did the authors obtain the hearts of the embryos and fetuses? It is very confusing and difficult to replicate.
- How was the stage of pregnancy ascertained for each rat, considering the number of rats per subgroup?
- “The diagnosis was confirmed” – please explain the diagnosis referred to.
- How were the animals euthanized?
- In Fig 1, the glucose levels were above 400 mg/dL from Day 9 through 17. Why say greater than 200 mg/dl and not 400 mg/dL?
Author Response
REVIEWER 2
1. How did the authors perform the vaginal smears? Briefly describe the procedure or cite the
appropriate reference(s).All changes to the manuscript text are showed in red.
Response: It has already been added in line 313.
2. How was the streptozotocin prepared?
Response: We used 50mg/kg of streptozotocin (P8833-100MG, Sigma Aldrich) diluted in 0.1M
sodium citrate solution (pH 4.5), always maintaining the cold environment. We cite the
appropriate reference, 11 and 29, in line 321.
Incomplete answer. How stable is streptozotocin in the buffer?
Response: Streptozotocin (STZ)-induced diabetes mellitus (DM) offers a very cost-effective
and rapid technique that can be used in most rodent strains. Manufacturer (Sigma) data sheets
indicate that STZ is stable for approximately 2 years if frozen (-20°C) and protected from the
light. In solution, STZ is stable around a pH of 4 and therefore is prepared in cold citrate
buffer at a pH of 4.5 to enhance stability. During the preparation of the STZ, we maintained
and ensured the cold network and the adequate pH, in addition, the maximum storage time
once prepared did not exceed one week.
References
- Motyl, K., & McCabe, L. R. (2009). Streptozotocin, type I diabetes severity and bone.
Biological procedures online, 11, 296–315. https://doi.org/10.1007/s12575-009-9000-5
- Ghasemi, A., & eddi, S. (2023). Streptozotocin as a tool for induction of rat models of
diabetes: a practical guide. EXCLI journal, 22, 274–294. https://doi.org/10.17179/excli2022-
5720
- Deeds, M. C., Anderson, J. M., Armstrong, A. S., Gastineau, D. A., Hiddinga, H. J.,
Jahangir, A., Eberhardt, N. L., & Kudva, Y. C. (2011). Single dose streptozotocin-induced
diabetes: considerations for study design in islet transplantation models. Laboratory
animals, 45(3), 131–140. https://doi.org/10.1258/la.2010.010090
Refer to the section on the Preparation of STZ Solution and the Summation in the second paper you
cited . Please make the appropriate correction.
3. How was the blood obtained to verify blood glucose levels and classify the rats as diabetic? Were
fasting blood samples used? If yes, how long was the fasting:
Response: The technique is mentioned in lines 326-327 and the type of blood samples was
corrected.
4. How many female rats were used for the study, and how many male rats?
Response: We used 12 female rats for the control group and 15 for the experimental group, in
addition to 8 male rats with the aim of completing a total of n=20 hearts per group and stage.
Not included in the revised manuscript. What was the ratio of males to females per cage?
Response: Two female rats were placed with one male per box, thank you for the observation,
the information is added in the text highlighted in blue.
5. When did the authors collect blood for glucose determination to confirm diabetes after
administering streptozotocin?
Response: This was carried out 48 hours after the induction with a tail blood sample, as
mentioned in line 326.
6. How did the authors obtain the hearts of the embryos and fetuses? It is very confusing and difficult
to replicate.
Response: We perform microdissections under a stereoscopic microscope using scissors and
forceps used in ophthalmology (added in line 333). We train and practice diligently until we
have mastered this skill and can replicate it to obtain the target organ.
7. How was the stage of pregnancy ascertained for each rat, considering the number of rats per
subgroup?
Response: The stage of pregnancy was determined after observing the presence of sperm in the
smear, counting this as day zero post-coitus.Was this done daily?
Response: It was done in the morning after the female's contact with the male (counting this as
day zero post-coitus).
8. “The diagnosis was confirmed” – please explain the diagnosis referred to.
Response: It has already been corrected in line 350.
9. How were the animals euthanized?
Response: Through active euthanasia with the administration of sodium pentobarbital and with
the support of expert veterinarians from our bioterium.
Not included in the revised manuscript.
Response: We have included the information in the text, highlighted in blue.
10. In Fig 1, the glucose levels were above 400 mg/dL from Day 9 through 17. Why say greater than
200 mg/dl and not 400 mg/dL?
Response: It was because 200 mg/dL was our inclusion criterion for the experimental group.
However, our induction method made us obtain even higher glucose levels.
Finally, we want to thank the reviewer for their valuable comments and constructive criticisms.
We have taken this into account in this manuscript and for the future.
Round 2
Reviewer 1 Report
Comments and Suggestions for Authors
The authors have revised their manuscript. Some concerns remain that the authors need to address.
1. To further introduce their findings on the mammallian heart (my earlier point 9) the authors should refer to studies by Nishino et al and Manivannan et al using STZ (PMIDs 39183978 and 35970860). The authors findings should also be discussed in the light of these studies.
2. The authors should note that the contribution of neural crest-derived cells to myocardium is controversial. AP2a expression is not a lineage marker and may simply be upregulated in outflow myocardium of treated embryos.
3. Typos in the reference list numbering (5, 31 and 27 are duplicated) need to be corrected.
Author Response
Response 1 and 2: We appreciate your support in strengthening the discussion with suggested articles. We have added the information highlighted in red.
Response 3: Thanks for the comment, the typos have already been corrected in the text.